# Relationship between Postural Stability, Lead Content, and Selected Parameters of Oxidative Stress

**DOI:** 10.3390/ijms232112768

**Published:** 2022-10-23

**Authors:** Marta Wąsik, Katarzyna Miśkiewicz-Orczyk, Michał Słota, Grażyna Lisowska, Aleksandra Kasperczyk, Francesco Bellanti, Michał Dobrakowski, Urszula Błaszczyk, Rafał Jakub Bułdak, Sławomir Kasperczyk

**Affiliations:** 1Department of Clinical Biochemistry and Laboratory Diagnostics, Institute of Medicine, Opole University, Oleska 48, 45-052 Opole, Poland; 2Department of Otorhinolaryngology and Laryngological Oncology, Medical University of Silesia in Katowice, Skłodowskiej-Curie 10, 41-840 Zabrze, Poland; 3ARKOP Sp. z o.o., Kolejowa 34a, 32-332 Bukowno, Poland; 4Department of Biochemistry, Faculty of Medical Sciences in Zabrze, Medical University of Silesia in Katowice, Jordana 19, 41-808 Zabrze, Poland; 5Department of Medical and Surgical Sciences, University of Foggia, Viale Pinto 1, 71122 Foggia, Italy

**Keywords:** posturography, lead, oxidative stress

## Abstract

This study attempts to determine whether the increased blood lead concentration affects the posturographic test and to determine the relationship between the parameters of posture stability and selected parameters of oxidative stress. The study population consisted of 268 male employees and was divided into two equal subgroups, depending on the lead content in the blood. A posturographic examination was performed. Concentrations of lead, cadmium, zinc protoporphyrin, selected essential elements, and selected markers of oxidative stress in the blood were tested. Higher blood lead concentrations positively affected the values of the sway results: the field and the mean velocity of the center of the feet pressure in posturography. The absolute value of the proprioception ratio was similar in both subgroups. The content of malondialdehyde shows a statistically significantly higher value in a subgroup with high blood lead concentration and exhibits significant correlations only with some of the posturography parameters. The lipofuscin content in erythrocytes correlates with the results of the posturography test. Zinc protoporphyrin, total oxidant status, total antioxidant capacity, selected minerals, and metals did not correlate with the results of the posturography test. In conclusion, posturographic results correlate only with selected markers of oxidative stress, so it can be assumed that the effect on the body balance is only partial.

## 1. Introduction

Lead is considered one of the most dangerous environmental toxic substances. Occupational exposure to lead compounds remains a serious problem, and its health effects have been extensively studied. Lead has no biochemical functions in the human body, and there is no safe level of this xenobiotic in the living and working environment. Exposure sources of lead include electronic waste, paint, batteries, glazed ceramics, cosmetics, traditional medicines, and industrial emissions in many countries all over the world [1].

Chronic lead poisoning usually manifests as negative changes in the functioning of many organs, although the typical intoxication symptoms involve gastrointestinal, neuromuscular, and neurological systems disorders. Lead primarily affects the central nervous system, particularly the developing brain. Within the brain, lead causes damage to the prefrontal cerebral cortex, hippocampus, and cerebellum. The effect of these damages can cause various neurological disorders, such as brain dysfunction, mental retardation, behavioral problems, nerve damage, and perchance Alzheimer’s disease, Parkinson’s disease, and schizophrenia [2]. Lead poisoning increases neurologic and psychiatric morbidity. Early symptoms of lead neurotoxicity include irritability, headaches, and impaired concentration. Chronic exposure to lead is neurotoxic to the human peripheral sensory system and has negative effects on cognitive functions [3].

The studies conducted so far indicate that lead also affects the renal system, impairs hematopoiesis, and is a causative agent of Pb-toxicity-related anemia, hypertension, as well as increased risk of stroke and cardiovascular disease incidence [4,5]. Environmental lead exposure has been associated with decreased kidney function, and lead nephrotoxicity is observed even at low levels of exposure [6]. Variant vitamin D receptor genes (VDR), polymorphisms of δ-aminolevulinic acid dehydratase (ALAD), and metallothioneins (MTs) expressions can also be altered following lead toxicity [7,8].

Exposure to lead compounds induces oxidative stress by generating reactive oxygen species (ROS) and dysfunction of antioxidant defenses, including enzymatic antioxidants, such as superoxide dismutase and catalase [9]. Short-term exposure to lead induces oxidative stress associated with elevated levels of lipid hydroperoxides (LPH) and affects the levels of essential minerals (such as calcium, magnesium, zinc, and copper) [10]. Competition between lead and magnesium for binding sites at the molecular and cellular levels, as well as at the systemic level, may represent an important aspect of lead toxicity in the context of oxidative stress, immune response, and gene expression modifications [11].

Even low blood lead levels (<10 μg/dL) are considered to be associated with deleterious effects on hematopoietic, nervous, and renal systems by inducing oxidative stress [4].

In a review article, Castellanos and Fuente described the negative auditory effects of heavy metals in populations exposed to lead [12]. Chronic exposure to lead causes damage to the balance organ, shared by the selected structures of the inner ear [13]. Min et al. findings hypothesized that blood lead and cadmium levels may be associated with balance and vestibular dysfunction in a general population of U.S. adults [14].

In the complex system of maintaining a body balance, an important role is played by the central processing and coordination of visual information coming from the periphery, from the vestibular organ of equilibrium and proprioreceptors of the skin, muscles, and tendons. The information transmitted from the central nervous system to the musculoskeletal system results in the stabilization of the center of the feet pressure [15]. Normal body posture coordination requires both somatosensory and visual information, and each joint movement and muscle control are uniquely influenced by a coordinated signal from multiple sensors [16]. Upright stability can be affected by neurotoxins (such as lead), so postural stability testing seems may be a suitable tool for detecting the early effects of toxins on the nervous system and determining the peripheral and central nervous system interactions. The computerized foot pressure center motion test is a non-invasive, fast, and reproducible method of testing for balance control used to test children, adults, and the elderly [17].

Posturography is a method of objective examination of balance and accurately differentiates disability status in people [18,19,20,21,22,23]. This method is a reliable and valid measure of impaired balance related to disorders of the central and peripheral parts of the vestibular organ. Exposure to lead at subclinical doses can cause adverse health effects that are not detected in a standard clinical trial. Therefore, subclinical health effects can be observed in occupationally exposed individuals without clinical symptoms or signs. Objective tests of the vestibular organ (such as peripheral nerve conduction velocities and balance testing) can be applied for the detection and documenting of subclinical changes before people exposed to lead develop clinical neurological disease. Neurophysiological studies are, therefore, a suitable research tool allowing for early detection and prevention of the development of neurological diseases.

The aim of this study was to determine whether the blood lead content affects postural stability and to determine the relationship between the parameters of posture stability and selected parameters of oxidative stress.

## 2. Results and Discussion

### 2.1. Demographic Values and Biochemical Test Results

The results of physical measurements, demographic results, and health habits are presented in Table 1. The research group was divided into two homogeneous subgroups: L-Pb (<33 µg/dL PbB) and H-Pb (≥33 µg/dL PbB). Subgroups was equal: *n* = 134 each. There were no significant differences in demographic data between subgroups.

The results of the blood chemistry test are presented in Table 2. The median lead content in the entire study group was 33.6 µg/dL PbB (twelve-year average value).

### 2.2. Posturography Test

The results of the posturography tests are presented in Table 3. For additional clarification, Table 4 and Table 5 contain a separate list of only the test results in sequence zero foam pads and with two foam pads under the feet. The results of posturography show dependence on the lead content between the subgroups; in particular, the statistical significance is noticed in the parameters: sway field F_COP_ and average velocity V_COP_.

Our study showed that the selected results of posturographic examination differ significantly in the H-Pb subgroup from the results in the L-Pb subgroup (Table 3). With a higher lead content in the blood, the following values of the sway indices were increased: the field covered by the COP (F_COP_) and the average velocity of the COP (V_COP_). Moreover, selected posturography results (in particular F_COP_ and V_COP_), especially in variants with eyes open, positively correlate with lead content in the entire research group (Table 5, below). This applies to both the average blood lead content: in the current value and also in the twelve-year average value. The current average lead content in L-Pb and H-Pb was slightly higher than the average from the previous 12 years (cumulative value) (Table 2). The nervous system, both central and peripheral, appears to be the most sensitive organ to lead toxicity. Even at a low level of exposure, negative effects on the nervous system may appear: loss of the myelin-insulating layer of neurons and reduced motor activity, which in turn weakens the nervous signal, weakens skeletal muscles and consequently causes a lack of muscle coordination [2].

In our study, the zinc protoporphyrin content was statistically significantly higher in the H-Pb than in the L-Pb (Table 2). This content was higher in the subgroup with the higher lead content, both in the most current sample taken and in the samples from the last 12 years (mean value). However, no statistically significant correlation was found between the ZPP concentration and the results of posturography (Table 5). ZPP is a commonly used marker of the influence of lead on the human body [24]. Lead inhibits the enzyme ferrochelatase, an enzyme involved in heme formation. The negative influence of lead on heme synthesis causes the production of zinc protoporphyrin and, as a result, the development of anemia. After entering the body, the half-life of Pb is approximately 40 days. Therefore, it is considered that the concentration of PbB may not be a sufficient parameter to assess the long-term effects of exposure to lead. An increase in ZPP in adults occurs when the blood lead content reaches the 30 μg/dL threshold [25].

The statistical analysis of posturography test results is presented in Table 4.

Our research showed that selected posturography results (in particular F_COP_ and V_COP_), especially with eyes open, positively correlate with lead content. In addition, posturography results positively correlate with age (in almost all parameters of the test variants) but only partially with work experience (Table 4). In body balance disorders, age progression should be taken into account because as the body ages, negative changes in the nervous system increase. However, the L-Pb and H-Pb research subgroups did not differ significantly in age: variation 4%, *p* = 0.123 (Table 1). Thus, the main negative causative factor here will not be the length of service itself (converted into the age of the respondent) but the number of years of work under exposure to lead compounds (marked in Table 1 as years of work exposure to Pb). Therefore, regardless of the age of the employee, work in occupational exposure to lead compounds carries the risk of negative consequences, including postural imbalances.

Selected posturography results in the H-Pb subgroup show a statistically significant increase in value compared to the L-Pb subgroup (Table 3). The increase in measured values concerns:-F_COP_—field covered by the center of feet pressure path plotted against time—statistical significance was demonstrated in all test variants (number of foam pads and eye status: 0o, 0c, 1o, 1c, 2o, 2c). The mean results of this parameter show a positive change between the L-Pb and H-Pb subgroups, ranging from 24% to 43% (calculations based on the values in Table 3). The mean values of this parameter show an increasing tendency with increasing difficulty of the test (Figure 1). Moreover, the upward trends (slope of the trend line) show similarity in the L-Pb and H-Pb subgroups in tests with different eye statuses: open/closed.-V_COP_—average velocity of the center of feet pressure displacement in the coronal or sagittal plane—the mean values of this parameter show statistical significance in almost all test variants (0o, 1o, 1c, 2o, 2c). The parameter mean values show a positive difference between the subgroups, ranging from 14% to 24% (calculations based on the values in Table 3). The mean values of this parameter show an increasing tendency with increasing difficulty of the test (Figure 2), and there is also an upward trend (slope of the trend line), which shows similarity in subgroups in tests with different eye statuses: open/closed.

The surface area covered by the center of feet pressure (F_COP_) and average velocity of the center of feet pressure (V_COP_) are presented in the figures below (Figure 1 and Figure 2).

The regression analysis showed that the results of the posturography test (zero foam pads, eyes open) positively correlated with the mean blood lead content (PbB current value, *p* < 0.0034).

The results of our posturographic study were consistent with previous studies on the effect of lead intoxication on body posture balance. There was a significant increase in the values of postural sway (with eyes closed) in a group with higher PbB concentrations in the parameters F_COP_ and V_COP_ in studies of workers chronically exposed to lead [26]. In the study of environmentally exposed children with very low lead exposure, statistically greater results were obtained in the subject with higher PbB in both variants: with eyes open and also with eyes closed [27].

The rest of the posturography parameters in our study showed a statistically significant positive difference between the L-Pb and H-Pb subgroups only in some variants of the test or did not show it at all. This indicates the diagnostic usefulness of only selected parameters of the sway test method, in particular: F_COP_ and V_COP_.

Our posturography results were in part in agreement with the swaying studies obtained by other researchers: Chia and others [28] report that the cumulative PbB (the past two years before the postural sway assessment) was significantly correlated with the postural sway parameters. On the other hand: the postural sway parameters (with eyes closed) were not significantly correlated with PbB or cumulative blood lead content. The complexity of the interactions between lead and posturography test outcomes may lead to divergent conclusions and thus require further research.

Our study of posturography was organized so that the subsequent variants allow the use of stimuli from different systems: vision, proprioreception, and vestibular stimuli. When the subjects are standing with open eyes on a hard surface (0o), all receptors (eyesight, proprioreceptors in tendons, and vestibular system) are involved in maintaining an upright body posture. Therefore, the sway field value for this condition is the smallest as all physiological afferent stimuli are available. In the eyes-closed version (0c), the visual sense signals are turned off, and the postural response reflects the availability of proprioreceptor signals and a slightly increased signal from the vestibular system. The sway area is greater than in previous test conditions and is dependent on signal processing by the central nervous system. The soft surface under the feet (foam pad) disrupts the signals from the proprioreceptors. Posture reflects a slightly increased input from the vestibular system and depends on how the vestibular organ is capable of compensating for and processing the changing signals coming from the feet. In the last variant of the test (2c), the response reflects abnormal signals from the proprioreceptors without the sense of sight available, and thus maintaining the balance is much more vestibular compared to earlier variants of the test. The sway field result for this test gives the highest values due to conflicting signals from the proprioreceptors and the vestibular system.

To better understand the role of different afferent stimuli in maintaining overall postural balance, physiological indicators were calculated. The importance of presenting data with these ratios effectively minimizes the possible effect of extraneous variables such as height and body weight. Based on the study by Bhattacharya [29], the appropriate indicators for F_COP_ were calculated (Figure 3):-Effect of the sense of sight on the results: vision ratio = 0c/0o;-Effect of change in proprioreception of lower extremities: proprioreception ratio = 2o/0o;-The effect of the greatest influence on the vestibular system (labyrinth in the inner ear and vestibular nuclei in the CNS): vestibular ratio = 2c/0o.

The ratios were calculated for both subgroups: L-Pb and H-Pb.

In this study, the absolute value of the proprioreception ratio was similar in both subgroups (Figure 3). The vestibular ratio was lower in H-Pb than in L-Pb. The obtained results were similar to previous studies [29]. However, the visual ratio was lower in H-Pb than in L-Pb. The obtained results of the analysis suggest a malfunction of the vestibular function. In the test with eyes closed, body posture is based on signals from the proprioreceptors and the vestibular system. Lead, by affecting the disorders of the central nervous system and disturbances of nerve conduction, may consequently cause imbalance. Thus, a reduced value of the vestibular ratio indicates a loss in the use of these afferents, resulting from the increased blood lead content in the H-Pb subgroup.

### 2.3. Oxidative Stress Markers

In our study, the content of malondialdehyde (MDA) shows a statistically significantly higher value in H-Pb than in L-Pb: by 13%, *p* = 0.043 (Table 2). MDA, a product of lipid peroxidation, has a cytotoxic, mutagenic, and carcinogenic effect, as well as the inhibition of the enzymes associated with the defense of cells against oxidative stress. MDA is one of the most frequently used indicators of lipid peroxidation and a biomarker for oxidative stress [30]. In this study, the results of posturography correlate with oxidative stress only at selected parameters. MDA correlates only with part of the posturography parameters: variants 0c and 2o. This indicates that oxidative stress is not the dominant mechanism in imbalance disorders. By including the dependencies of lead concentration and posturography results, it is more likely a direct negative influence of lead on the pathogenesis of equilibrium disturbances.

In this study, the lipofuscin content in red blood cells does not depend on the concentration of lead: LPS content in the H-Pb subgroup was higher by 3% but without statistical significance (Table 2). However, the lipofuscin content in erythrocytes correlates with the results of the posturography test (Table 4). Lipofuscin is an unwanted intracellular component that is highly resistant to proteolytic degradation. Reactive oxygen species initiate lipid peroxidation through the production of MDA, which affects the cross-linking of macromolecules such as proteins and lipid peroxidation products [31]. This causes the formation of lipofuscin aggregates, consisting mainly of oxidized proteins and lipids. The above results may indicate an influence on posturography both through an increase in MDA content and through oxidative stress leading to an increase in the amount of lipofuscin.

The oxidative stress marker TAC indicated a statistically significantly higher value in the subgroup with a higher lead content (by 2%, *p* = 0.043) than in the subgroup with a lower content of this heavy metal in the blood (Table 2). Lead and its compounds interact covalently with the sulfhydryl groups of glutathione and with enzymes that participate in its redox transformations, causing their inactivation. Lead also inhibits other important antioxidant enzymes (superoxide dismutase, catalase), which results in the lack of excessive removal of oxygen free radicals [32]. In our study, TOS, TAC, and OSI did not correlate with the results of the posturography test. This suggests that maintaining proper body balance is only partially dependent on oxidative stress. Thus, the sense of balance is partly influenced by the direct negative properties of lead and partly by increased oxidative stress. The results obtained in our study require further studies to more precisely define the role of oxidative stress in balance disorders.

### 2.4. Minerals and Metals: Cd, Ca, Fe, Mg, Zn

The results of our study show that the content of cadmium is statistically significantly higher in the H-Pb subgroup (by 26%, *p* = 0.040) than in the subgroup with a lower content of this heavy metal (Table 2). However, in the Spearman rank correlation, it was not found that the cadmium content influenced posturographic values. In this study, the dominance of lead influence does not allow to define the role of cadmium in a negative effect on body posture.

The calcium content is statistically significantly lower in the subgroup with high lead concentration (by 3%, *p* = 0.011) than in the subgroup with low lead concentration (Table 2). This is consistent with previous studies, in which workers chronically exposed to lead showed a negative correlation between the level of lead in the blood and the concentration of Ca in plasma [10,33]. This result is confirmed by the inhibitory mechanism of lead to 1α-hydroxylase in the renal tubules, where, as a consequence, calcitriol synthesis is inhibited, resulting in a reduction in calcium absorption in the intestine and its reabsorption in the renal tubules [34]. However, no correlation was found between the calcium content and posturographic values (Spearman rank correlation).

A wide range of experimental evidence proves the important role of iron in the diet in cases of lead poisoning [35]. Iron deficiency may be associated with higher lead uptake and subsequent negative effects of lead poisoning [36]. In our study, appropriate blood tests have been performed to eliminate that iron deficiency can affect the results. In the subgroup with higher blood lead concentrations, the Fe content was 5% higher (no statistical significance) than in the second subgroup. In the Spearman rank correlation, it was not found that the iron concentration affected posturographic values.

Magnesium deficiency transiently leads to pro-oxidant effects and increases oxidative stress by contributing to the increased production of reactive oxygen species, increasing the sensitivity of tissues to oxidative damage [37]. A trace element: zinc has a great influence on the body’s homeostasis, immune function, oxidative stress, and aging. Zn has antioxidant properties, reduces lead-induced oxidative stress, and also competes with lead for similar binding sites [38]. In our study, the content of magnesium and zinc showed lower values in the group with H-Pb (Table 2), by 2% and 1%, respectively (no statistical significance). It was found that the concentrations of magnesium or zinc were not statistically correlated with the results of the posturography test.

### 2.5. Summary, Strengths, and Limitations

In summary, the results obtained in this study are presented in detail in Table 5, divided into:-Strong results—lists the results obtained, highlighting the most important of them;-Results requiring further research—setting out possible directions of projects in the future.

The strengths and limitations of this study are presented in Table 6.

## 3. Material and Methods

### 3.1. Study Population

The research group consisted of 268 male employees of the lead-zinc smelter in Miasteczko Śląskie, Poland. The smelter produces zinc and lead by pyrometallurgical extraction, which is based on the reduction of roasted Zn-Pb concentrate with coke at 1000 °C in a shaft furnace. All of the subjects working at various positions in the steelworks (dust removal operator, electro-automation worker, capacitor worker, controller, fitter, heavy equipment operator, locksmith, machine operator, mechanic, rectification operator, refiner, shaft furnace operator, sinter plant operator, smelter, welder) were exposed to lead and cadmium compounds. Data on age, weight, height, working time under exposure to lead and cadmium, medical history, and smoking habit were obtained from the personal questionnaire.

The inclusion criterion for the classification of the subjects into the research group was occupational exposure to Pb and PbB ≥ 20 μg/dL. Work experience was at least 5 years. Exclusion criteria included a medical history of any chronic disease (without diabetes, coronary artery disease, hypertension), abnormal physical examination, symptoms, and signs of any infectious disease, or cancerous disease. People with otolaryngological diseases (past and present) were also excluded from the study.

Since there is no normative value for lead, the study population was arbitrarily divided into two subgroups based on the median level of the Pb in the blood (33 µg/dL): L-Pb (low-PbB-level subgroup, PbB < 33 µg/dL, *n* = 134) and H-Pb (high-PbB-level subgroup, PbB ≥ 33 µg/dL, *n* = 134).

### 3.2. Posturography Examination

A posturography test was performed with the aim to assess the center of feet pressure (COP). The test consisted of changing the direction of tilting the center of pressure exerted by the feet: in the anterior-posterior (AP) and lateral plane (L), on a stable surface, and on a sponge. The study was performed on a CATSYS 2000—SWAY 7.0 static posturography (Danish Product Development Ltd., Snekkersten, Denmark) with a single-plate platform.

The testing protocol was based on that of Bhattacharya [29], with subsequent modifications. The subjects stood on a stable platform, without shoes, possibly still, with their arms along the body, looking straight ahead. The inclinations of the body were recorded in six successive variants, each lasting 30 s, of increasing difficulty.

A test with two foam pads placed under the feet translates into an increased degree of difficulty compared to the previous version of the test with one foam pad. The assessed parameters of the posturographic examination and the characteristics of the various test variants are presented in Table 7.

### 3.3. Laboratory Procedures

Blood samples were collected from a peripheral vein, using sterile test tubes coated with K_3_EDTA (Vacuette; Greiner-Bio, Frickenhausen, Germany) (to obtain whole blood) or plain tubes (to obtain serum). Serum was collected using centrifugation aliquoted, and stored at −80 °C until analysis.

### 3.4. Concentrations of Lead (Pb), Cadmium (Cd), and Zinc Protoporphyrin (ZPP)

Blood levels of lead (PbB) and zinc protoporphyrin (ZPP) were applied as biomarkers of lead exposure. The cadmium exposure was estimated from the cadmium concentration in the blood. Assessments of the blood lead concentration (PbB) and blood cadmium concentration (CdB) were conducted by graphite furnace atomic absorption spectrometry technique using the iCE 3400 AAS Spectrometer (Thermo Fisher Scientific, Waltham, MA, USA). The obtained results were expressed as μg/dL for the Pb level and μg/L for the Cd level. The blood concentration of zinc protoporphyrin (ZPP) was measured directly using Aviv Hematofluorometer HF Model 206 (Aviv Biomedical, Lakewood, NJ, USA), using an excitation wavelength of 415 nm and an emission wavelength of 596 nm. The instrument measures the ratio of ZPP fluorescence to the reference sample absorption (hemoglobin alone), displayed as μg ZPP per g of hemoglobin (μg/g Hb).

### 3.5. Concentrations of Selected Essential Elements: Iron (Fe), Calcium (Ca), Magnesium (Mg), and Zinc (Zn)

The assessments of the serum concentrations of iron, calcium, magnesium, and zinc were performed using graphite furnace atomic absorption spectrometry in an iCE 3400 instrument (Thermo Fisher Scientific, Waltham, MA, USA). The following wavelengths were used: iron: 248.3 nm, calcium: 422.7 nm, magnesium: 285.2 nm, and zinc: 213.9 nm. The results were expressed in μg/dL for the Fe serum concentration and mmol/L for the Ca, Mg, and Zn serum concentration.

### 3.6. Biochemical Procedures of Antioxidant Defense and Oxidative Stress Markers

Malondialdehyde concentration (MDA) was measured fluorometrically as a 2-thiobarbituric acid-reactive substance (TBARS) in serum, according to Ohkawa et al. [39], with modifications. Samples were mixed with sodium dodecyl sulfate (8.1%), acetic acid (20%), and 2-thiobarbituric acid (0.8%), then vortexing. Incubation lasted 1 h at 95 °C, and butanol-pyridine 15:1 (*v*/*v*) solution was added. The samples were shaken (10 min) and centrifuged. The butanol-pyridine layer was measured fluorometrically (515 nm and 522 nm wavelengths) (PerkinElmer, Waltham, MA, USA). TBARS values are expressed as malondialdehyde (MDA) equivalents. Tetraethoxypropane was used as the standard. Concentrations were expressed in serum: μmol/L and in erythrocytes nmol/g of hemoglobin (Hb).

The lipofuscin (LPS) concentration in the samples was determined according to the Jain method [40]. Fluorescence was measured spectrofluorimetrically at 360 nm (absorbance) and 440 nm (emission) (LS45, PerkinElmer, Waltham, MA, USA). The results are presented as relative lipid extract fluorescence (RF) in erythrocytes: RF/g of hemoglobin (Hb). The value of 100 RF corresponds to the fluorescence of a solution of 0.1 µg/mL quinidine sulfate in 0.05 mol/L (0.1 N) sulfuric acid.

Total oxidant status (TOS) in serum was measured according to Erel [41]. The method is based on the oxidation of ferrous ions (Fe^2+^) to ferric ions (Fe^3+^) in oxidizing agents in an acidic medium. The color change of xylenol orange occurred as a result of the reaction. The change in absorbance was measured at a wavelength of 560 nm. The data were expressed as μmol/L.

Total antioxidant capacity (TAC) in serum was measured according to Erel [42]. The method is based on the synthesis of free radicals, which are reduced by antioxidants contained in the tested samples. The result of the process is a color change of the ABTS+ ion (2,2′-azino-bis (3-ethyl-benzothiazoline-6-sulfonic acid) diammonium salt). The results were expressed as mmol/L.

The oxidative stress index (OSI) was calculated as the ratio of TOS to TAC. The results were expressed as a percentage value.

### 3.7. Time Sequence of the Study

The study was time-extensive and covered a period of up to 12 years back (see Figure 4).

Based on the patient files, the results of tests up to 12 years ago were obtained: the results of the PbB and ZPP concentrations in the blood, physical examination results, and demographic data of employees. Workers routinely underwent a series of preventive examinations due to the fact that they worked under exposure to lead, noise, and other factors.

In the last year of the project, a current biochemical blood test was performed, including markers of oxidative stress, and the group was examined with a posturographic test, and the results of the body balance were obtained. The current results of the physical examination, demographic data, and the most recent blood chemistry (the content of Cd and other elements in the blood) were also collected.

### 3.8. Statistical Analysis

The statistical analysis was performed using the STATISTICA 13 PL software (StatSoft). Spearman’s rank correlation analysis was performed to determine possible relationships between lead concentrations and various questionnaire variables, as well as relationships between rocking parameters. The regression was analyzed, and the level of statistical significance was set at *p* < 0.05.

## 4. Conclusions

Selected posturography results, especially with eyes open variants, positively correlate with lead content. Selected posturography results in the subgroup with higher lead content in the blood show a statistically significant increase in value compared to the subgroup with a lower content of this heavy metal in the blood. This indicates the importance of the negative influence on balance disorders by lead concentration through the influence on the central nervous system and nerve conduction disturbances.

Posturographic results correlate only with selected markers of oxidative stress, so it does not prove the existence of a dominant mechanism in this case. The proper body balance is only partially affected by oxidative stress. The results obtained in our study require further research aimed at a more precise determination of the role of oxidative stress in body balance disorders.

## Figures and Tables

**Figure 1 ijms-23-12768-f001:**
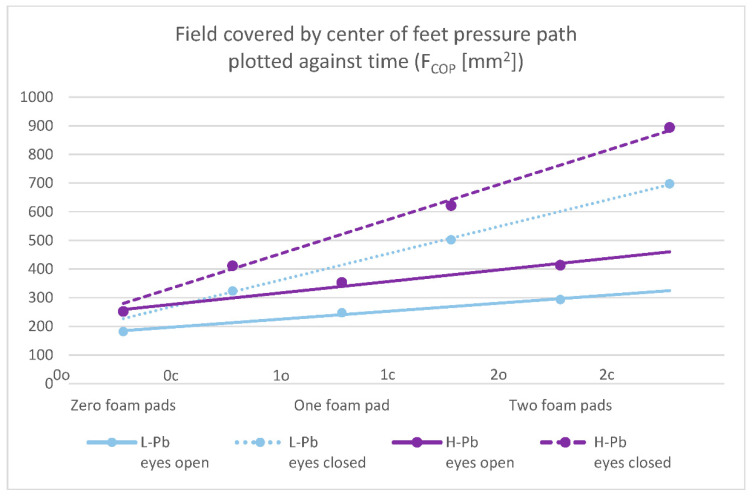
Posturography test results: a field covered by the center of feet pressure path plotted against time—F_COP_ (mm^2^). L-Pb—low-PbB-level subgroup, H-Pb—high-PbB-level subgroup. Posturographic parameters are described in Table 7. A trend line has been marked. Calculations based on the values in Table 3. The level of statistical significance was set at *p* < 0.05.

**Figure 2 ijms-23-12768-f002:**
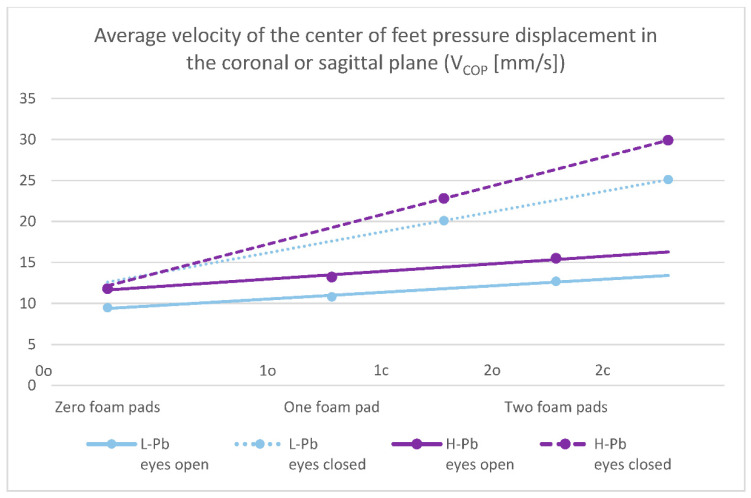
Posturography test results: average velocity of the center of feet pressure displacement in the coronal or sagittal plane—V_COP_ (mm/s). L-Pb—low-PbB-level subgroup, H-Pb—high-PbB-level subgroup. Posturographic parameters are described in Table 7. A trend line has been marked. Calculations based on the values in Table 3. The graph shows only statistically significant results (*p* < 0.05).

**Figure 3 ijms-23-12768-f003:**
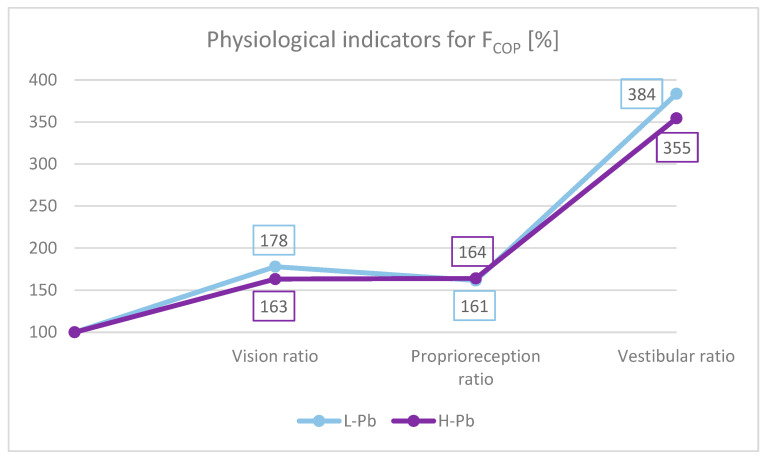
F_COP_ indicators: vision ratio, proprioreception ratio and vestibular ratio. L-Pb—low-PbB-level subgroup, H-Pb—high-PbB-level subgroup. The chart shows the values of the indicators in percent. Calculations based on the values in Table 3.

**Figure 4 ijms-23-12768-f004:**
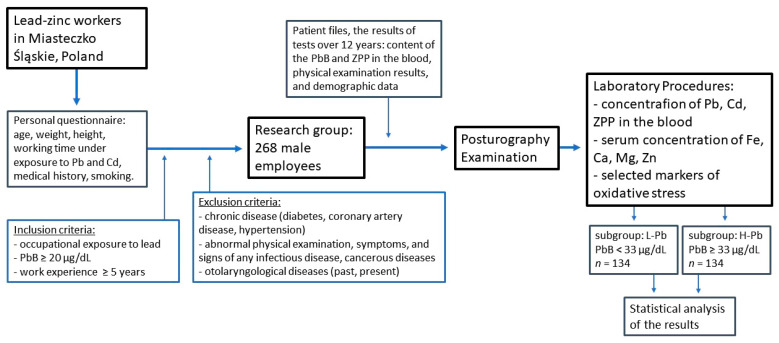
Flowchart of the study of the relationship between postural stability, blood lead concentration, and selected parameters of oxidative stress. ZPP—zinc protoporphyrin concentration, PbB—blood lead concentration, L-Pb—low-PbB-level subgroup, H-Pb—high-PbB-level subgroup.

**Table 1 ijms-23-12768-t001:** Descriptive statistics for physical measurements and health habits.

		L-Pb PbB < 33 (µg/dL) *n* = 134		H-Pb PbB ≥ 33 (µg/dL) *n* = 134		Variation (%)	*p*-Value
		Mean	SD/n	Mean	SD/n		
Age (years)		38.91	9.83	40.57	7.53	4%	0.123
Years of work in exposure to Pb (years)		10.63	9.27	13.78	7.29	30%	0.002
Body weight (kg)		84.62	12.63	88.11	14.74	4.1%	0.083
Height (cm)		177.17	6.80	177.52	5.76	0.2%	0.713
BMI		26.70	4.58	27.91	4.13	4.5%	0.062
DM		1.9%	5	3.5%	3		0.503
CAD		1.9%	5	1.2%	1		0.679
HA		9.7%	27	9.4%	7		0.945
Smoking	Yes/No	22.5%	63	34.2%	26		0.084
	In the past (years)	13.2	7.6	13.5	7.7	2.4%	0.832
	Currently (number of pieces)	13.2	6.5	11.1	5.8	−15.9%	0.231

The level of statistical significance was set at *p* < 0.05, marked black font. BMI—body mass index, DM—diabetes (*Diabetes mellitus*), CAD—coronary artery disease, HA—hypertension (*Hypertonia arterialis*), L-Pb—low-PbB-level subgroup, H-Pb—high-PbB-level subgroup.

**Table 2 ijms-23-12768-t002:** Biochemical test results.

	L-Pb PbB < 33 (µg/dL) *n* = 134		H-Pb PbB ≥ 33 (µg/dL) *n* = 134		Variation (%)	*p*-Value
	Mean	SD/n	Mean	SD/n		
PbB (µg/dL) *	23.0	10.3	41.5	7.6	80%	<0.001
PbB (µg/dL) **	22.9	8.7	41.1	4.5	80%	<0.001
ZPP (µg/g Hb) *	3.80	1.80	7.07	3.70	86%	<0.001
ZPP (µg/g Hb) **	3.68	1.60	6.66	2.82	81%	<0.001
CdB (µg/L)	1.99	1.70	2.51	1.95	26%	0.040
Ca (mmol/L)	2.45	0.26	2.39	0.26	−3%	0.011
Fe (µg/dL)	20.9	7.2	22.0	7.9	5%	0.665
Mg (mmol/L)	0.82	0.15	0.81	0.11	−2%	0.460
Zn (mmol/L)	14.1	4.5	13.9	4.7	−1%	0.862
MDA in serum (µmol/L)	2.67	1.52	3.01	1.02	13%	0.043
MDA in erythrocytes (nmol/g Hb)	242.7	79.4	227.0	71.1	−6%	0.109
LPS (RF/g Hb)	665.1	340.9	683.5	293.2	3%	0.657
TOS (µmol/L)	10.5	4.0	10.2	3.8	−3%	0.605
TAC (mmol/L)	1.12	0.13	1.14	0.11	2%	0.043
OSI (%)	0.97	0.45	0.92	0.43	−5%	0.385

The level of statistical significance was set at *p* < 0.05, marked black font. PbB—blood lead concentration, *—current value, **—twelve-year average value, ZPP—zinc protoporphyrin concentration, CdB—blood cadmium concentration, MDA—malondialdehyde concentration, LPS—lipofuscin concentration, TOS—total oxidant status, TAC—total antioxidant capacity, OSI—oxidative stress index, L-Pb—low-PbB-level subgroup, H-Pb—high-PbB-level subgroup.

**Table 3 ijms-23-12768-t003:** Posturography test results.

	L-Pb PbB < 33 (µg/dL) *n* = 134		H-Pb PbB ≥ 33 (µg/dL) *n* = 134		Variation (%)	*p*-Value
	Mean	SD/n	Mean	SD/n		
Zero foam pads, eyes open (0o):						
L_COP_ (mm)	4.55	1.86	5.02	1.78	10%	0.041
MAPS_COP_ (mm)	2.59	1.23	2.88	1.00	11%	0.040
MLS_COP_ (mm)	3.19	1.42	3.49	1.49	9%	0.096
F_COP_ (mm^2^)	181.8	171.6	252.1	209.0	39%	0.003
V_COP_ (mm/s)	9.48	3.46	11.79	8.53	24%	0.005
S_inte_ (mm/s)	4.44	1.75	4.81	1.77	8%	0.310
Zero foam pads, eyes closed (0c):						
L_COP_ (mm)	5.68	2.21	6.12	1.93	8%	0.086
MAPS_COP_ (mm)	3.38	1.52	3.67	1.32	8%	0.113
MLS_COP_ (mm)	4.10	2.89	4.18	1.51	2%	0.795
F_COP_ (mm^2^)	323.5	300.4	411.3	302.1	27%	0.020
V_COP_ (mm/s)	17.5	24.2	18.2	8.1	4%	0.758
S_inte_ (mm/s)	6.08	2.11	6.59	2.14	8%	0.054
Two foam pads, eyes open (2o):						
L_COP_ (mm)	5.71	2.08	6.11	2.18	7%	0.136
MAPS_CO_ (mm)	3.41	1.36	3.63	1.42	6%	0.202
MLS_COP_ (mm)	3.82	1.82	4.21	1.65	10%	0.070
F_COP_ (mm^2^)	293.6	247.4	413.2	330.4	41%	0.001
V_COP_ (mm/s)	12.7	5.1	15.5	6.6	22%	<0.001
S_inte_ (mm/s)	5.58	1.95	6.41	2.50	15%	0.003
Two foam pads, eyes closed (2c):						
L_COP_ (mm)	8.33	2.90	9.68	8.70	16%	0.097
MAPS_COP_ (mm)	5.14	2.00	5.58	1.85	9%	0.068
MLS_COP_ (mm)	5.44	2.05	5.80	1.86	7%	0.139
F_COP_ (mm^2^)	697.4	556.8	893.7	595.8	28%	0.007
V_COP_ (mm/s)	25.1	11.5	29.9	12.3	19%	0.001
S_inte_ (mm/s)	8.84	3.16	10.35	9.34	17%	0.029

The level of statistical significance was set at *p* < 0.05, marked black font. Posturographic parameters are described in Table 7. L-Pb—low-PbB-level subgroup, H-Pb—high-PbB-level subgroup.

**Table 4 ijms-23-12768-t004:** Posturography test—Spearman’s rank order correlation, *p* < 0.05, marked black font.

	Age (Years)	Years of Work in Exp. to Pb (Years)	PbB (µg/dL) *	PbB (µg/dL) **	ZPP (µg/g Hb) *	ZPP (µg/g Hb) **	MDA in Serum (µmol/L)	MDA in Eryth. (nmol/g Hb)	LPS (RF/g Hb)	TAC (mmol/L)
Zero foam pads, eyes open (0o):										
L_COP_ (mm)	0.20	0.16	0.19	0.17	0.08	0.05	0.04	0.06	0.16	−0.06
MAPS_COP_ (mm)	0.15	0.13	0.18	0.17	0.04	0.03	0.07	0.05	0.18	−0.07
MLS_COP_ (mm)	0.19	0.15	0.17	0.15	0.09	0.06	0.04	0.06	0.12	−0.05
F_COP_ (mm^2^)	0.17	0.16	0.23	0.22	0.14	0.13	0.06	0.02	0.15	−0.05
V_COP_ (mm/s)	0.13	0.17	0.13	0.18	0.11	0.07	0.05	−0.02	0.08	0.03
Zero foam pads, eyes closed (0c):										
L_COP_ (mm)	0.23	0.18	0.12	0.14	0.10	0.09	0.15	0.02	0.13	−0.03
MAPS_COP_ (mm)	0.24	0.21	0.13	0.15	0.09	0.11	0.14	−0.03	0.11	−0.06
MLS_COP_ (mm)	0.19	0.14	0.10	0.13	0.10	0.08	0.12	0.03	0.12	0.00
F_COP_ (mm^2^)	0.24	0.20	0.19	0.22	0.12	0.13	0.13	0.00	0.14	−0.06
V_COP_ (mm/s)	0.21	0.21	0.12	0.16	0.11	0.10	0.00	−0.01	0.05	−0.02
Two foam pads, eyes open (2o):										
L_COP_ (mm)	0.16	0.07	0.11	0.11	0.02	0.04	0.10	−0.04	0.17	−0.04
MAPS_COP_ (mm)	0.11	0.05	0.11	0.10	0.00	0.02	0.13	0.02	0.11	−0.02
MLS_COP_ (mm)	0.16	0.04	0.11	0.11	0.04	0.05	0.04	−0.10	0.12	−0.03
F_COP_ (mm^2^)	0.15	0.12	0.21	0.20	0.10	0.12	0.15	−0.02	0.13	−0.05
V_COP_ (mm/s)	0.17	0.15	0.17	0.21	0.10	0.11	0.14	−0.01	0.12	0.04
Two foam pads, eyes closed (2c):										
L_COP_ (mm)	0.16	0.09	0.14	0.18	0.04	0.05	0.02	0.07	0.08	−0.06
MAPS_COP_ (mm)	0.11	0.07	0.11	0.15	−0.02	0.01	−0.01	0.08	0.15	−0.02
MLS_COP_ (mm)	0.19	0.11	0.15	0.18	0.11	0.09	0.03	0.04	0.01	−0.06
F_COP_ (mm^2^)	0.17	0.15	0.18	0.22	0.05	0.10	0.09	0.11	0.20	−0.06
V_COP_ (mm/s)	0.20	0.16	0.16	0.23	0.10	0.13	0.10	0.08	0.18	−0.03

Posturographic parameters are described in Table 7. PbB—blood lead concentration, *—current value, **—twelve-year average value, ZPP—zinc protoporphyrin concentration, MDA—malondialdehyde concentration, LPS—lipofuscin concentration, TAC—total antioxidant capacity.

**Table 5 ijms-23-12768-t005:** Summary of the results.

Strong Results	Results Requiring Further Research
With higher blood lead content, the following sway index values increased: COP area (F_COP_) and average COP speed (V_COP_). Selected results of posturography (F_COP_ and V_COP_), especially with eyes open, positively correlate with the content of PbB in the entire research group. This applies both to the average PbB: in the present value as well as in the twelve-year average. Selected posturography results in the H-Pb subgroup show a statistically significant increase in value compared to the L-Pb subgroup. Posturography results, especially with eyes open, positively correlate with age (in almost all parameters of the test variants) but only partially with work experience. The absolute value of the proprioreception ratio was similar in both subgroups. The vestibular ratio was lower in H-Pb than in L-Pb. The ZPP content was statistically significantly higher in the H-Pb than in the L-Pb subgroup. The content of MDA shows a statistically significantly higher value in H-Pb than in L-Pb. The LPS content in erythrocytes correlates with the results of the posturography test. The TAC shows a statistically significantly higher value in the H-Pb subgroup than in the L-Pb subgroup.	Posturography correlates with oxidative stress only at a selected few parameters of oxidative stress. MDA correlates only with part of the posturography parameters. The LPS content in red blood cells does not depend on the concentration of lead. TAC, TOS, and OSI did not correlate with the results of the posturography test. The content of Cd was statistically significantly higher in the H-Pb subgroup than in the L-Pb subgroup. However, there was no statistically significant correlation between the cadmium content and the posturographic values. The Ca content was statistically significantly lower in the H-Pb subgroup than in the L-Pb subgroup. However, no correlation was found between the calcium content and posturographic values. In the H-Pb subgroup, the Fe content was higher (no statistical significance) than in the second subgroup. In the Spearman rank correlation, it was not found that the iron content influenced posturographic values. The content of Mg and Zn showed lower values in the H-Pb subgroup (no statistical significance). There was no correlation between the concentrations of Mg or Zn and the results of the posturography test.

COP—center of feet pressure, PbB—blood lead concentration, L-Pb—low-PbB-level subgroup, H-Pb—high-PbB-level subgroup, ZPP—Zinc protoporphyrin concentration, MDA—malondialdehyde concentration, LPS—lipofuscin concentration, TAC—total antioxidant capacity, TOS—total oxidant status, OSI—oxidative stress index.

**Table 6 ijms-23-12768-t006:** Strengths and limitations of this research.

Strengths	Limitations
A large research group (268 people). A homogeneous research group (age, sex, body weight, height, chronic diseases, health behavior). Current biochemical data and twelve-year average value. The objective of the project was clearly stated. The inclusion and exclusion criteria were stated. A flow chart with a study design has been included. The laboratory procedures used were appropriate. The author stated that there was no conflict of interest.	Presence of confounding factors: influence of other heavy metals in the workplace (Cd). No control group (steel workers not exposed to lead compounds). Limited possibility of generalization (specificity of working conditions).

**Table 7 ijms-23-12768-t007:** Characteristics of the parameters and the test variants of the posturographic examination.

Parameter	Description
**Posturographic Examination**	
L_COP_	Path length—the total distance traveled by the COP in the specified time, described in mm
MAPS_COP_	Mean sway of the COP from point 0 in the anterior-posterior direction, measured in mm
MLS_COP_	Mean sway of the COP from point 0 in the lateral direction, in mm
F_COP_	Field covered by the COP path plotted against time, described in mm^2^
V_COP_	Average velocity of the COP displacement in the coronal or sagittal plane, in mm/s
S_inte_	Sway intensity—the root mean square of accelerations, recorded in the 0.1 Hz to 10.1 Hz band during the test period, in mm/s
SI	Sway index *—factors specified by the manufacturer
**Test Variants**	
0o	Zero foam pads (standing directly on the platform), eyes open
0c	Zero foam pads, eyes closed
1o	One foam pad under the feet, eyes open
1c	One foam pad, eyes closed
2o	Two foam pads, eyes open
2c	Two foam pads, eyes closed

COP—center of feet pressure, *—SI was not included in this study.

## Data Availability

Not applicable.

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
