# Peer review of "Relationship between Postural Stability, Lead Content, and Selected Parameters of Oxidative Stress"

_ijms, 2022, doi:10.3390/ijms232112768_

Round 1
Reviewer 1 Report
The manuscript entitled:” Relationship between postural stability, lead content and selected parameters of oxidative stress” aimed to evaluate the correlation of lead levels with postural stability problems and with oxidative stress generated by lead exposure. The manuscript is well written and of interest to the research community. Some improvements are needed before the manuscript be accepted.
1. A flowchart that presents the study design should be included in the material and methods.
2. Strengths and limitations of the study should be included in the discussion section.
Author Response
Reply to the Review Report
Thank you for all your comments. The advice to add a project flowchart made it very clear the outline of the research project and the Material and Methods chapter.
Point 1: A flowchart that presents the study design should be included in the material and methods.
Response 1: A block diagram showing the study design has been provided (section 2. Material and Methods, subsection 2.7. Time Sequence of the Study, Figure 1. - file attached). The new numbering of figures has been taken into account.
Point 2: Strengths and limitations of the study should be included in the discussion section.
Response 2: An additional table is provided at the end of the discussion: strengths and limitations of this study (section 3. Results and Discussion, subsection 3.5. Strengths and limitations of this study, Table 7 - file attached).
There were also minor corrections and typing errors that had escaped when checking the previous version of the manuscript.

Reviewer 2 Report
Dear Editors,
Thank you for giving me the opportunity to review the article entitled: Relationship between postural stability, lead content and selected parameters of oxidative stress
The study aimed to determine whether the blood lead content affects postural stability, and to determine the relationship between the parameters of posture stability and selected parameters of oxidative stress. The study population consisted of 268 male employees and was divided into two equal subgroups, depending on the lead content in the blood. A posturographic examination was performed.
The study population consisted of 268 male employees and was divided into two equal subgroups, depending on the lead content in the blood. A posturographic examination was performed. Concentrations of lead, cadmium, zinc protoporphyrin, selected essential elements and selected markers of oxidative stress in the blood was tested. Higher blood lead concentrations positively influenced the values of the sway results: the field and the mean velocity of the center of the feet pressure in posturography.
The authors concluded that selected posturography results, especially with eyes open variants, positively correlate with lead content. In the H-Pb subgroup shows a statistically significant increase in value compared to the L-Pb subgroup. This indicates the importance of the negative influence of lead concentration on balance disorders. Posturographic results correlate only with selected markers of oxidative stress, so it does not show a dominant mechanism in this case. The proper body balance is only partially affected by oxidative stress.
The following suggestions are pointed out:
· In the introduction section: page 2 line 19: …… inducing oxidative stress [4].
page 2 line 22: Min et al. findings suggested …...
· In the Materials and Methods section: The subtitles could be revised.
· The title of Table 1 could be revised.
· The conclusion should reflect that the obtained results require further studies to define the role of oxidative stress more precisely in balance disorders.
The manuscript ijms-1958498 is suitable for publication in IJMS , Section Molecular Toxicology Special Issue Medical and Environmental Aspects of Metal Toxicity and could be accepted after minor revision for the above-mentioned suggestions.
Author Response
Reply to the Review Report
Thank you for your comments, they helped to improve the text of the manuscript. All the corrections allowed to improve the description of the methods, the conclusions are clearly presented and more supported by the results.
Point 1: In the introduction section:
page 2 line 19: …… inducing oxidative stress [4].
page 2 line 22: Min et al. findings suggested…….
Response 1: Omitted formatting of the citation source record has been corrected, and text improvements have been applied.
Point 2: In the Materials and Methods section: The subtitles could be revised.
Response 2: As recommended, the subtitles in the materials and methods section have been redrafted: we’ve changed "2.2. Posturography Study" to "2.2. Posturography Examination", "2.5. Biochemical Procedures of Selected Essential Elements Concentrations" to 2.5. "Concentrations of Selected Essential Elements: Iron (Fe), Calcium (Ca), Magnesium (Mg), and Zinc (Zn)".
Point 3: The title of Table 1 could be revised.
Response 3: The title of table 2 (new numbering) has been changed to make the title more relevant to its content: we’ve changed "Table 1. Demographic results" to " Table 2. Descriptive statistic for physical measurements and health habits" (section 3. Results and Discussion, subsection 3.1. Demographic values and biochemical test results).
Point 4: The conclusion should reflect that the obtained results require further studies to define the role of oxidative stress more precisely in balance disorders.
Response 4: The conclusions were supplemented with the missing information (section 5. Conclusions):
“The results obtained in our study require further research aimed at more precise determination of the role of oxidative stress in body balance disorders.”
And minor corrections were made and typing errors that had escaped during the review of the previous version of the manuscript were corrected.
Reviewer 3 Report
The authors try to relate blood lead content in workers with postural stability and some parameters of oxidative stress. This research work implies a great effort. Congratulations to the authors. Nevertheless, before publication, some points need to be addressed:
Minor:
- In introduction correct [4 Flora et al. 2012].
- Reference to previous works done by other authors should be done wither in introduction or results and discussion. Example of other works:
https://pubmed.ncbi.nlm.nih.gov/16216016/
https://pubmed.ncbi.nlm.nih.gov/10810103/
- Description of parameters in section 2.2 should be done in a table for better readability.
- Space should be given between text and the legends of figures and tables.
Major:
- Authors should comment the result for the relation between years of work and lead content. In this sense, a clarification is needed why are not the postural differences related simply to higher years of work?
- For better interpretation of results and comprehension of the research novelty, the results must be written together with the discussion of results in a single chapter (Results and Discussion).
- Table 4 is confusing and should be improved.
- Figure 1, 2 and 3 should be mentioned in the results and not only in the discussion. These figures should also be aligned with results that are presented in the table.
- In conclusions the authors are too generalist and should be more specific. For example similar to what is said in the discussion “Selected posturography results (in particular FCOP and VCOP), especially with eyes open, positively correlate with lead content. Posturography correlates with oxidative stress only at a selected, few parameters of oxidative stress: mainly with lipofuscin (LPS). Malondialdehyde in serum (MDA) correlates partially, only in some variants of the posturographic test. Also posturography results, especially with eyes open, positively correlate with age.”
Author Response
Reply to the Review Report
Thank you for your comments and advice, they helped to improve the text of the manuscript, making it clearer. The conclusions we want to show in our study are clearer and more supported by the results.
Point 1: In introduction correct [4 Flora et al. 2012].
Response 1: In the introduction, the formatting of the citation source record has been corrected: [4].
Point 2: Reference to previous works done by other authors should be done wither in introduction or results and discussion. Example of other works:
https://pubmed.ncbi.nlm.nih.gov/16216016/
https://pubmed.ncbi.nlm.nih.gov/10810103/
Response 2: The reference to previous works, in line with the examples provided, was moved to one of the chapters: introduction (page 2, lines 37 to 46). The numbering in the reference list was updated, which has changed after this operation.
Point 3: Description of parameters in section 2.2 should be done in a table for better readability
Response 3: As recommended, the characteristics of the posturographic examination parameters and test variants are described in table 1, to make the use of abbreviated names of parameters and reading their descriptions more efficient (section 2. Material and Methods, subsection 2.2. Posturography Examination, Table 1. Characteristics of the parameters and the test variants of the posturographic examination - file attached). As a consequence, the numbering of further tables in the content of the manuscript has been updated.
Point 4: Space should be given between text and the legends of figures and tables.
Response 4: One distance line has been added after the legend under the tables and figures to separate this part of the text from the content of the right article, which visually improved the arrangement of manuscript elements.
Point 5: Authors should comment the result for the relation between years of work and lead content. In this sense, a clarification is needed why are not the postural differences related simply to higher years of work?
Response 5: To clarify the relationship between working years and lead content, the statement in Table 2 on "working years in exposure to Pb" has been clarified. In addition, the sentence in the paragraph was expanded (page 8, lines 33 to 43):
“Our research showed that selected posturography results (in particular FCOP and VCOP), especially with eyes open, positively correlate with lead content. Also posturography results positively correlate with age (in almost all parameters of the test variants), but only partially with work experience (Table 5). In body balance disorders, age progression should be taken into account, be-cause as the body ages, negative changes in the nervous system increase. However, the L-Pb and H-Pb research subgroups did not differ significantly in age: variation 4%, p=0.123 (Table 2). Thus, the main negative causative factor here will not be the length of service itself (converted into the age of the respondent), but the number of years of work exposed to lead compounds (marked in Table 2 as years of work in exposure to Pb). Therefore, regardless of the age of the employee, work in occupational exposure to lead compounds carries the risk of negative consequences, including postural imbalances.”
Point 6: For better interpretation of results and comprehension of the research novelty, the results must be written together with the discussion of results in a single chapter (Results and Discussion).
Response 6: The results were rearranged and combined and discussed into one chapter: 3. Results and Discussion. This, as recommended, improved the interpretation of the results, strengthened the support of the discussed conclusions with the results obtained, improved their clarity and presentation. The numbering list of the cited literature has been updated, which has changed after this operation.
Point 7: Table 4 is confusing and should be improved.
Response 7: In the Table 4 (previously), now number 5 (file attached), has been corrected: redundant descriptions from the lines (on the left) have been removed, replaced with the description of the entire variant (top, middle), misleading description regarding statistical significance has been removed, descriptions too long from the top of the columns (Current value, Twelve-year average value) has been replaced with “*” and “**” characters, respectively, their meaning is expanded in the legend below the table. The introduced corrections improved the readability of the table and ensured that it will no longer cause problems.
Point 8: Figure 1, 2 and 3 should be mentioned in the results and not only in the discussion. These figures should also be aligned with results that are presented in the table.
Response 8: The listed figures are now in the Results and Discussion chapter, references in the text have been added, and it has been ensured that the figure has information about the reference to the table (where the input data for the calculations was taken from).
Point 9: In conclusions the authors are too generalist and should be more specific. For example similar to what is said in the discussion “Selected posturography results (in particular FCOP and VCOP), especially with eyes open, positively correlate with lead content. Posturography correlates with oxidative stress only at a selected, few parameters of oxidative stress: mainly with lipofuscin (LPS). Malondialdehyde in serum (MDA) correlates partially, only in some variants of the posturographic test. Also posturography results, especially with eyes open, positively correlate with age.”
Response 9: Instead of repeating elements of the discussion in the conclusions and extending the division of conclusions excessively, we allowed ourselves to summarize the results in the form of a table. (subchapter 3.5. Summary, strengths and limitations, Table 6 - file attached). The key results achieved are highlighted in detail in "Strong results". On the right, the weaknesses of this study are listed and indicate a possible direction for further research in the future. The conclusions were reformulated by adding information about the need for further research to better investigate the influence of oxidative stress on the balance of human body posture.
And minor word corrections were made, spelling mistakes were removed which were accidentally omitted when checking a previous version of the manuscript.

Round 2
Reviewer 1 Report
The authors addressed all my comments. The manuscript is much improved compared with the initial version so my recommendation is accept.
Reviewer 3 Report
The authors greatly improved the manuscript. Consider to used "Solid Results" instead of "Strong Results" in table 6.